# Dual Responsive poly(vinyl caprolactam)-Based Nanogels for Tunable Intracellular Doxorubicin Delivery in Cancer Cells

**DOI:** 10.3390/pharmaceutics14040852

**Published:** 2022-04-13

**Authors:** Kummara Madhusudana Rao, Maduru Suneetha, Dachuru Vinay Kumar, Hyeon Jin Kim, Yong Joo Seok, Sung Soo Han

**Affiliations:** 1School of Chemical Engineering, Yeungnam University, 280 Daehak-ro, Gyeongsan 38541, Gyeongbuk, Korea; msraochem@yu.ac.kr (K.M.R.); msunithachem@gmail.com (M.S.); hyeonjin3438@ynu.ac.kr (H.J.K.); dydwn106@gmail.com (Y.J.S.); 2Research Institute of Cell Culture, Yeungnam University, 280 Daehak-ro, Gyeongsan 38541, Gyeongbuk, Korea; 3Research Center for Herbal Convergence on Liver Disease, College of Korean Medicine, Daegu Haany University, Gyeongsan 38610, Gyeongbuk, Korea; dachurivinay@gmail.com

**Keywords:** poly(vinyl caprolactam), nanogels, dual responsive, drug release, intracellular triggered, cancer cells

## Abstract

In this work, doxorubicin (Dox)-encapsulated poly(vinyl caprolactam) (PVCL)-based three-dimensional nanogel networks were developed and were crosslinked with disulfide linkages. The nanogels degrade rapidly to low molecular weight chains in the presence of the typical intracellular concentration of glutathione. Doxorubicin (Dox) was successfully encapsulated into these nanogels. The nanogels have a high drug loading of 49% and can be tailored to 182 nm to deliver themselves to the targeted cells and release Dox under dual stimuli conditions, such as redox and temperature. By evaluating cell viability in the HepG2 cell line, we observed that Dox-loaded nanogels effectively killed the cancer cell. Fluorescence microscopy results show that the nanogels could easily be internalized with HepG2 cells. The results confirm that the nanogels destabilized in intracellular cytosol via degradation of disulfide bonds in nanogels networks and release of the Dox nearby the nucleus. These carriers could be promising for cancer drug delivery.

## 1. Introduction

Cancer is one of the leading causes of death in humans. It is defined mainly by uncontrolled cell proliferation and the capacity of cells to spread [1]. Despite tremendous progress in the fight against cancer, it continues to be a difficult medical problem, especially in the lungs, breast, liver, prostate, pancreas, and brain [2]. To date, systemic chemotherapy has been widely used to treat cancer patients on a long-term basis [3]. On the other hand, systemic toxicity is a significant drawback that limits the usability and effectiveness of chemotherapeutics. Recent research efforts in drug delivery systems have focused on targeted delivery and controlled release of the drug or other agents in the tumor [4]. By designing an effective drug delivery system (DDS) with controlled drug release at a specific site, the drug concentration will be maintained in the optimal therapeutic range with a single dose. The study of cancer-targeted drug delivery utilizing polymeric nanoparticles has developed significantly in recent years [5,6,7]. There is increasing confidence that nanotechnology used for drug delivery may substantially improve cancer treatment. However, most studies utilizing nontherapeutic formulations such as micelles and liposomes fail to show therapeutic benefits in cancer treatment due to their toxic side effects [8]. Due to their easy tailoring features and ability to successfully encapsulate anticancer drugs of different types through simple methods and respond to external stimuli based on their functional polymeric networks, nanogels are being investigated as drug delivery agents targeting cancer [9,10,11]. Nanogels can be selectively internalized by cancer cells and avoid accumulating in non-target tissues, leading to lower therapeutic doses and fewer harmful side effects [9,10,11,12]. Nanogels as a drug delivery platform can improve cancer chemotherapy effectiveness and help cancer patients.

The design of intracellular degradable nanocarriers offers an efficient order to avoid the premature release of drug molecules at the extracellular level than in the intracellular environment. The fabrication of nanocarriers makes such intracellular triggered smart design possible via ultrasound, temperature, magnetic field, light, enzyme, pH, and redox potential [13]. Among these, the employment of degradable nanocarriers to initiate on-demand drug release is important to address the challenge mentioned above. The intracellular stimuli conditions of cancer cells are entirely different from normal cells and the extracellular environment. Recently, redox-responsive self-assembled and metal-organic framework nanocarriers have been developed for targeted drug delivery in cancer cells [14,15,16,17]. Other studies have reported that the addition of disulfide linkers on shell crosslinking and interlayer crosslinking enhances the drug release in cancer cells [18,19]. These nanocarriers degrade into the original carbon–carbon bond-based polymer chain when disulfide cleavage occurs on reduction. As a result, the extent of degradation is limited to long polymer chains. Integrating disulfide bonds into polymeric nanoarchitecture is a challenge that enables extended degrees of degradation and programmable drug release. To do so, further possibilities of stimuli-sensitive nanogels via one-step synthesis must be explored. A disulfide functionalized cross-linker produces stable structures for efficient drug delivery to cancerous cells rather than extracellular conditions.

Temperature-responsive nanocarriers can change their phase transition behavior, swollen at lower critical solution temperature changes (LCST) due to the hydrogen between water and the amide functional group of polymers, and shrinking at higher critical solution temperatures (LCST) due to hydrophobic interactions [20]. The feature of LCST behavior can tune the drug release to avoid the side effects of therapeutic agents. PVCL (poly(vinyl caprolactam)) is a thermoresponsive polymer with an LCST (32–34 °C) similar to PNIPAM (poly(N-isopropyl acrylamide) [20]. PVCL research is still in its infancy, especially in comparison to PNIPAM. PVCL is also biocompatible, which makes it more desirable to the field of biomedicine [21]. PVCL, on the other hand, has yet to receive FDA approval, despite the increasing number of reports highlighting its potential in drug delivery, tissue engineering, and regenerative medicine. In biomedical fields, including drug delivery, many micro/nano gel studies have been successfully employed [17,18,19,20]. Although various functional groups in stimuli-responsive PVCL nanogels have been reported, new synthetic methods to modify them with multiple functional groups for cancer-targeted drug delivery may also be needed. Hydrophilic polymers play an important role in improving the LCST of thermoresponsive polymers. Hydroxyethyl acrylate (HEA) is a hydrophilic monomer have been used to prepare hydrogels for tissue engineering and drug delivery applications, because it can improve the hydrophilicity and elasticity of hydrogel [22].

By considering the above-mentioned facts, we developed new redox degradable three-dimensional nanogels of P(VCL-HEA) that have disulfide in the networks for loading Dox as an anticancer drug. To the best our knowledge, there are no reports on the cytosol degradable nanogels based on P(NVCL-co-HEA) nanogels having disulfide linkages in the network for Dox drug delivery. The incorporation of HEA into nanogel structure becomes more hydrophilic, and the consequent overall hydrogen bonding ability of polymer chains leads to a higher phase-transition temperature. The nanogels that were produced with Dox were stable at extracellular conditions can preferentially release an anticancer drug with degradation of network disulfide linkages in response to the intracellular acid conditions. In addition, cytotoxicity studies also evaluated drug-loaded nanogels on HepG2 cancer cells.

## 2. Materials and Methods

### 2.1. Materials

N-vinyl caprolactam (VCL), hydroxyethyl acrylate (HEA), N,N′-Bis(acryloyl)cystamine (Cys-BIS), 2,2′-Azobis(2-methylpropionitrile) (AIBN), sodium dodecyl sulfate (SDS), and 1,4-dithiothreitol (DTT) were purchased from Sigma-Aldrich Co. Ltd, Seoul, Korea. VCL was dissolved in hexane, purified using a short alumina column, and then recrystallized before the polymerization process. Recrystallization of AIBN using methanol was also used to purify it.

### 2.2. Preparation of Cys-BIS-P(VCL-HEA) Nanogels

The batch emulsion polymerization method was employed in a round bottom flask with a reflux condenser and a nitrogen gas inlet to prepare Cys-P(VCL-HEA) NGs. Briefly, in 100 mL of double-distilled water (DDW), SDS (40 mg), and AIBN (50 mg predessolved in methanol solution) were dissolved, and then VCL (1.8 g), HEA (0.4 g), and Cys-BIS (4 wt%) were added to the SDS solution. The mixture was bubbled with nitrogen gas for 45 min, then heated to 70 °C and vigorously stirred for 6 h at 800 rpm. The mixture was taken out at the end of the reaction, cooled to room temperature, and dialyzed against water (MWCO = 3.5 kDa) for two weeks to eliminate unreacted monomers. Finally, the obtained Cys-BIS-P(VCL-HEA) nanogels dispersion was lyophilized for two days and stored for further analysis.

### 2.3. Characterization

The crystalline nature of Dox and Dox-loaded Cys-BIS-P(VCL-HEA) nanogels was evaluated using an X-ray diffraction technique (2θ= 10–50; XRD, Bruker AXS D8 advance, CuKa radiation source (λ = 1.54)) at a scan speed of 5/min and working at 40 kV and 30 mA. A UV-visible spectrophotometer was used to record the ultraviolet-visible (UV-Visible) absorption spectra of the Dox loading samples (UV-Vis; Shimadzu-2600). Dynamic light scattering (DLS) on a Malvern Zetasizer Nano-ZS was used to determine the particle size and distribution of Cys-BIS-P(VCL-HEA) nanogels (DLS; Malvern Instrument). Transmission electron microscopy (TEM, JEOL JEM-2010) was used to examine the size and shape of Cys-BIS-P(VCL-HEA) nanogels at a 200 kV accelerating voltage. A drop of nanogels (dispersed in DDW) was dropped on the surface of a copper grid and dried under a lamp.

### 2.4. Dox Loading

Briefly, 5 mg/2.5 mL of Dox (2 mg/mL) was added to a 5 mL DDW containing 100 mg ultrasonically well-dispersed Cys-BIS-P(VCL-HEA) nanogels, and the mixture was stirred for 24 h at room temperature in the dark. Dox-loaded Cys-BIS-P(VCL-HEA) nanogels were centrifuged (10,000 rpm, 10 min) and washed with DDW to remove physically bound Dox on the surface of Cys-BIS-P(VCL-HEA) nanogels. The supernatant was collected and stored in the dark to measure unloaded Dox using UV-Visible spectrophotometry at a fixed wavelength (480 nm).

### 2.5. In Vitro Dox Release from Cys-BIS-P(VCL-HEA)

Five milligrams of Dox-loaded Cys-BIS-P(VCL-HEA) was incubated with 10 mL of PBS (pH 7.4 conditions) in a shaker with 100 rpm at 37 °C. The samples were centrifuged, the supernatant was collected, and the nanogels were resuspended at different time intervals (1, 2, 4, 8, 12, and 24 h). The effect of temperature (25 °C and 37 °C) on Dox release from nanogels was also studied using the same protocol. To check redox-triggered Dox release from nanogels, the Dox-loaded Cys-BIS-P(VCL-HEA) nanogels were treated with different concentrations of DTT with PBS (5 mM and 10 mM). The released amount of Dox was determined using UV-Visible spectra at a fixed wavelength (480 nm).

### 2.6. Cell Culture

HepG2 (hepatocellular carcinoma) and CCDK-normal skin fibroblasts (American Type Culture Collection) cells were grown in Dulbecco’s Modified Eagle Medium (DMEM), which included 10% fetal bovine serum (FBS) and 100 IU/mL penicillin at a concentration of 100 mg/mL. The cells were maintained at 37 °C in a humidified incubator with a 5% CO_2_ environment.

### 2.7. Cytotoxicity Study

Cells were trypsinized manually by counting directly using a hemocytometer. The cells were seeded into 96 well tissue culture plates at a density 5000 to 15,000 cells/cm^2^ and allowed to attach cells using the appropriate cell culture medium (100 μL) for 24 h. Cell culture media was removed, followed by the addition of samples. For CCDK cells, pristine Cys-BIS-P(VCL-HEA) nanogels suspension at different concentrations were added. For HepG2 cells, Dox and Dox-loaded Cys-BIS-P(VCL-HEA) nanogels with different concentrations of Dox were added. After 72 h of incubation, the toxicities were determined with Prastoblue^®^ (Invitrogen by thermos fisher scientific, Eugene, OR, USA) Cell Viability Assay. Finally, the absorbance of each well was measured with wavelengths of 570 and 600 nm. All assays were conducted in triplicate. The mean values and their standard error of the means were calculated.

### 2.8. Cell Uptake and Intracellular Distribution of Dox

Cells (HepG2) at a density 5000 cells/cm^2^ were seeded onto cover glass slips and incubated for 24 h. Then, 2.5 μg/mL of free Dox and Dox-loaded Cys-BIS-P(VCL-HEA) nanogels were added to each well and incubated for 3 h. Afterward, 500 μL of 4% formaldehyde solutions were added in to each well incubated for 10 min. The cells were counterstained with DAPI for 20 min. The wells were washed with PBS (pH = 7.4), and the glass coverslips were carefully removed from each well and mounted onto glass slides using Fluoromount as vectorshield (Sigma Aldrich, Seoul, Korea). The uptake was confirmed with fluorescence microscopy.

## 3. Results and Discussion

### 3.1. Preparation and Characterization of Dox-Loaded Cys-BIS-P(VCL-HEA) Nanogels

In this study, we designed 3D-network nanogels with disulfide linkages. The Cys-BIS integrated P(VCL-HEA) nanogels were made using a solution polymerization process. To create 3D-network nanogels in SDS solution, two important monomers, VCL and HEA, as well as Cys-BIS as a crosslinker, were employed. The overall formation of Cys-BIS integrated P(VCL-HEA) nanogels is depicted in Figure 1. The amphiphilic SDS molecules might delay nucleation and hinder nanogel particle size growth. The monomers and crosslinker acrylic functional group are initiated by AIBN, producing stable 3D networks. Furthermore, the nanogels are strengthened by the H-bonding interaction between the VCL and HEA functional groups. This method allows the formation of a uniform size and shape of nanogels.

Dox is widely employed in the treatment of different types of tumors, as this molecule can bind DNA and block the synthesis of biomacromolecules [23]. To better understand the potential applications of the obtained nanogels for drug delivery, Dox was selected as a model drug and used in the study of loading and release properties of the nanogels. The fabricated nanogels were then loaded with Dox using equilibrium swelling in Dox solutions. As shown in Figure 1a, the UV-Vis spectrometry of Dox solutions before and after loading to the nanogels represents the decrease in absorbance of Dox. The Dox content (wt%) was calculated as 49%. Due to hydrophilic functionality and 3D-network structure, the Dox could easily be defused into nanogels networks and then stabilized through the formation of hydrogen-bonding interactions between Dox and nanogels functional groups (Figure 1). Furthermore, XRD patterns were studied to know the physical state of Dox in the Cys-BIS-P(VCL-HEA) nanogels networks (Figure 1b,c). Pure Dox shows many characteristic XRD peaks, while nanogels are amorphous (Figure 1b) [24]. The Dox crystalline peaks were not observed in drug-loaded Cys-BIS-P(VCL-HEA) nanogels, suggesting that Dox crystalline character turns into an amorphous state and that Dox is distributed evenly in the nanogels matrix.

The shape, size, and morphology of nanogels can affect drug loading and cellular internalization to treat cancer cells [25]. Understanding the behavior of drug delivery systems requires accurate shape, size, and morphology of nanogels. This will assist in determining their utility as drug delivery vehicles. DLS studies reveal that the highly controllable colloidal stable nanogels was observed (Figure 2a). The average diameter of the Cys-BIS-P(VCL-HEA) nanogels is found to be narrow (polydispersity is 0.196), and the average diameter of the nanogels is 182 nm. It is sufficient to internalize nanogels in the cancer cells, which can promote the in vivo fate of a colloidal drug delivery system. 

The morphology of the Cys-BIS- P(VCL-HEA) was investigated by TEM (Figure 2b–d). It was found that the Cys-BIS-P(VCL-HEA) had spherical morphology with a high uniform size. In this study, the particle size of the nanogels produced approximately 123 nm. The reduction in particle size over the DLS study is due to the dehydration of the nanogels that occurred during the TEM sample preparation process.

### 3.2. Temperature and Redox-Responsive Nature of Cys-BIS-P(VCL-HEA) Nanogels

DLS was used to test the temperature responsiveness of Cys-BIS-P(VCL-HEA) (Figure 3a,b). When the temperature was elevated from 15 °C to 45 °C, a decrease in the hydrodynamic diameter of nanogels was noticed, resulting in a decrease in the diameter (i.e., 59% shrinkage). Between 32 °C and 34 °C, the fastest size transition occurred. The hydration–dehydration mechanism may readily understand such temperature responsiveness of PVCL chains [24]. The nanogel is in its hydrophilic condition at low temperatures, well-hydrated with trapped water via hydrogen bonding, but, at high temperatures, the nanogel collapses into its hydrophobic state, causing hydrogen bonds with water molecules to be disrupted, resulting in dehydration of the nanogel [24].

Many cancer cells overexpress the glutathione in cytosol environment. The concentration of glutathione levels in cancer cells is between 5–10 mM [25]. In general, disulfide bonds are cleavable under redox conditions. To prove the redox-responsiveness of developed nanogels, the nanogels were treated with DTT. The redox-responsive degradation of nanogels was observed in TEM and DLS studies. The nanogels were exposed to 5 mM and 10 mM of DTT reagent for 6 h at 37 °C. The treated samples were observed in the DLS study (Figure 3e). The results show that irregular size distribution was obtained after exposing nanogels to 5 mM DTT for 6 h because of the degradation behavior of nanogels under a redox environment. At lower 5 mM DTT and 37 °C, the nanogels are degraded into the low molecular weight of P(VCL-HEA) copolymer, which produces micellar aggregates due to the LCST behavior of PVCL chains. However, the structure of micellar structures is irregular. Furthermore, the nanogels were treated with 10 mM of DTT, which resulted in a narrow size distribution of nanogels micellar aggregates with <5 nm in size being observed, which indicates the complete degradation of nanogels. TEM images also proved the irregular shape of nanogels after being treated with 5 mM and 10 mM of DTT (Figure 3c,d). Hence, the disassembly behavior is mainly caused by the cleavage of crosslinked disulfide bonds under DTT conditions.

### 3.3. In Vitro Drug Release of Dox-Loaded Cys-BIS-P(VCL-HEA) Nanogels

Dox release from Dox-loaded Cys-BIS-P(VCL-HEA) nanogels was studied at two different temperatures (25 °C and 37 °C) in pH 7.4. As from Figure 4, the percent cumulative release of Dox from Cys-BIS-P(VCL-HEA) nanogels increased at a lower temperature (below LCST) of 25 °C, but decreased at a higher temperature (above LCST) of 37 °C (Figure 1). This is due to the rapid hydration of nanogel networks that are completely swollen at low temperatures. The polymer network structure collapses at high temperatures, resulting in a decreased ability to take water or buffer solution, reducing the drug diffusion rate from the nanogel networks. This result demonstrates that the nanogels respond to temperature. The results confirmed the developed nanogels are stable under extracellular conditions (pH 7.4 at 37 °C). The Dox delivery in intracellular conditions is a big problem. For this, the developed nanogels were integrated with disulfide linkages to trigger fast drug release in the intracellular cytosol environment, as many cancer cells overexpress glutathione levels, which can easily break the disulfide bonds (Figure 1). The Dox release was monitored at pH 7.4 with 5 mM and 10 mM DTT conditions. The prepared nanogels are highly stable to pH 7.4 at 37 °C, as discussed in their stability behavior from DLS studies. Hence, the release of Dox from nanogels under pH 7.4 is very low. At 5 mM DTT, more than 80% Dox was released within 24 h; notably, the DTT concentration change from 5 to 10 mM corresponds to the intracellular tumor tissue. Furthermore, the DTT concentration switch to 10 mM improved the release performance of the Dox from nanogels, with 95% Dox release within 24 h. It was concluded that the crosslinked disulfide bonds play the main role in the degradation of nanogels under intracellular redox stimuli trigger conditions.

### 3.4. In Vitro Cytotoxicity

The nanogels were incubated with CCDK-skin fibroblasts, and the biocompatibility of the prepared Cys-BIS-P(VCL-HEA) nanogels was evaluated using the Prestoblue viability assay. As shown in Figure 5a, the pure Cys-BIS-P(VCL-HEA) nanogels displayed good cytocompatibility even at high concentrations, with cell viability above 95% after 72 h of incubation.

Furthermore, the in vitro cytotoxicity of Dox-loaded Cys-BIS-P(VCL-HEA) nanogels was tested utilizing HepG2 cells (liver cancer cell line) and the Prestoblue assay to test the antitumor activity of Dox upon release from the nanogels Figure 5b. From the results, it has been observed that improved cytotoxicity of Dox-loaded Cys-BIS-P(VCL-HEA) nanogels incubated with 10 mM glutathione treated HepG2 cancer cells as compared to free Dox, Dox-loaded Cys-BIS-P(VCL-HEA) nanogels, and Cys-BIS-P(VCL-HEA) nanogels. The results suggest that Cys-BIS-P(VCL-HEA) nanogels effectively inhibited the HepG2 cancer cells with the half-maximal inhibitory concentration (IC50) is about 0.3529 μg/mL, which was lower than free Dox (0.7152 μg/mL).

### 3.5. Cellular Internalization of Dox-Loaded Cys-BIS-P(VCL-HEA) Nanogels

The intracellular uptake of the Dox-loaded Cys-BIS-P(VCL-HEA) nanogels was evaluated by fluorescence microscopy. The cellular uptake of Dox from the Dox-loaded Cys-BIS-P(VCL-HEA) nanogels was analyzed with the fluorescence of Dox (red) (Figure 5c). DAPI was regarded as a fluorescence marker for the visualization of the HepG2 cell nuclei. The results of cellular uptake after 3 h of incubation with the free Dox, Dox-loaded Cys-BIS-P(VCL-HEA) nanogel-treated HepG2 cells, are shown in Figure 5c. Red spots (DOX) were observed in the HepG2 cells, indicating that free Dox was observed in the nucleus [26]. The fluorescence intensity of Dox around the nucleus of HepG2 was observed for Dox-loaded Cys-BIS-P(VCL-HEA) nanogels, indicating the nanogels are easily internalized inside the HepG2 cancer cells. Furthermore, the intracellular triggered release of Dox was monitored within the HepG2 cancer cell by treating cells with the Dox-loaded Cys-BIS-P(VCL-HEA) nanogels. Their fluorescence intensity was observed around the nucleus and inside the nucleus, showing that the intracellular overexpressed HepG2 cells can easily destroy the nanogel networks inside the cancer cell, thereby releasing Dox nearby the nucleus [25]. Therefore, the developed Dox-loaded Cys-BIS-P(VCL-HEA) nanogels have promising intracellular redox responsive properties and are suitable for delivering bioactive agents to cancer cells.

## 4. Conclusions

In conclusion, we developed intracellular triggered Cys-BIS-P(VCL-HEA) nanogels for cancer drug delivery. The Cys-BIS-P(VCL-HEA) are biocompatible with CCDK-normal skin fibroblasts cells. The Cys-BIS-P(VCL-HEA) nanogels can load a maximum amount of Dox (49%) and are stable at extracellular conditions (pH 7.4 at 37 °C) due to the temperature-responsive property of nanogels. With GTH as a trigger, the model drug Dox could be released from the nanogels via degradation of disulfide bonds. In vitro results confirmed the nanogels could facilitate the internalization inside HepG2 cancer cells and trigger Dox release around the nucleus owing to redox-active disulfide bonds that existed in the nanogel networks, as well as the uniform size and shape of nanogels. Overall, the size- and shape-controlled disulfide-based nanogels may be a promising anticancer drug-releasing platform for facilitated on-demand drug release-pinpointing cancer chemotherapy with a much-improved safety profile, and this study may pave the way for developing other novel redox-sensitive nanogel formulations with targeting ligand functionalization for targeted drug delivery to the cancer therapy. 

## Data Availability

All available data are reported in the article.

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
