# Peer review of "Dual Responsive poly(vinyl caprolactam)-Based Nanogels for Tunable Intracellular Doxorubicin Delivery in Cancer Cells"

_pharmaceutics, 2022, doi:10.3390/pharmaceutics14040852_

Round 1

Reviewer 1 Report

The manuscript untitled: “Dual responsive poly(vinyl caprolactam) based nanogels for tunable intracellular doxorubicin delivery in cancer cells” is a very interesting and complete study about the preparation, physicochemical characterization, drug loading and release from nanogels, and drug uptake from nanogels by hepatocarcinoma cell lines. I recommend the publication of the manuscript in the presente form, because in my opinion all the importante parameters were evaluated in the study of the nanogels. It will be good to perform in vivo studies in an animal model.

Author Response

The authors show doxorubicin (Dox) encapsulated poly(vinyl caprolactam) (PVCL) based three dimensional nanogels networks were developed and were crosslinked with disulfide linkages, and this formulation were evaluated as characterization, Dox release, cytotoxicity and cellura uptake test. There are some points of interest in these results, but the reviewer conclude not to accept this manuscript for Pharmaceutics, because it is a short of research novelty. The authors described the PVCL (poly(vinyl caprolactam)) is still in its infancy, especially in comparison to PNIPAM (poly(N-isopropyl acrylamide), but there are some similar articles about the  nanoparticles including drug for cancer therapy. I feel the authors should emphasize the novelty of this article and the system discussed in this manuscript is superior to the other nanoparticle system. Moreover, the reviewer thinks that the results are not well discussed and conclusions are not meaningful in manuscript.

The major points are described as below.

In Figure 1 legend, (c) is lost.

Response: Thank you for your comment. Now we have provided figure legend.

Considering the order of the structure of the text, Figure 2(d) should be renumbered as Figure 2(a).

Response: Thank you for your comment. Now we have renumbered Figure. 2.

Please check the Figure 3 legend again. Figure and description do not match.

Response: Thank you for your comment. Now we have provided figure legend and description.

In Figure 5 legend, (a) and (c) are lost.

Response: Thank you for your comment. Now we have provided figure legend.

Page 9, line 327, Fig 6B and C were described, but there is no Fig. 6!

Response: Thank you for your comment. Now we have corrected.

Did the authors have the release profiles of DOX at 5 and/or 10 mM DTT  at 25 degree C? The reviewer thinks the faster release profiles 5 and/or 10 mM DTT  at 25 degree C would show rather than 37 degree C.

Response: Thank you for your comment. The authors did not study the release profiles of Dox at 25oC because of consideration of body temperature (37oC).

In vitro cytotoxicity,  the authors discussed to Cys-BIS-P(VCL-HEA) nanogels effectively inhibited the HepG2 cancer cells with the IC50 value, but cell viability seems to be almost the same between the 10 mM GTH group and DOX alone at DOX concentrations of 1 mg/mL or higher, since the tolerated dose of DOX in HepG2 cells and the actual concentration of  anticancer activity are unknown. It cannot be said that this formulation is more safety rather than Dox alone from this result.

Response: Thank you for your comment. The author’s apology for wrong description of cytotoxicity analysis. We assume that the treatment of GSH with HepG2 cells improves the redox levels in cells. As per reviewer 3 comment, we have repeated experiment without GSH treatment of HepG2 cells. As per literature, HepG2 cells overexpress intracellular GSH levels. We repeat the experiment without GSH. The results confirmed the cell viability of Dox and Dox loaded nanogels are not similar. Now we have revised.

In cellular uptake test, I do not recognized the more Dox fluorescence at 10 mM of GTH intensity around the nucleus and inside the nucleus than 5 mM GTH from Fig. 5(C), the authors should add a quantitative test or a test with a different cell system or fluorescent dye.

Response: Thank you for your comment. As per Fluorescence images we could confirm the dox is internalized inside cancer cells and nucleus. In future, we will perform other cell lines and in vivo study.  

Reviewer 2 Report

The authors show doxorubicin (Dox) encapsulated poly(vinyl caprolactam) (PVCL) based three dimensional nanogels networks were developed and were crosslinked with disulfide linkages, and this formulation were evaluated as characterization, Dox release, cytotoxicity and cellura uptake test. There are some points of interest in these results, but the reviewer conclude not to accept this manuscript for Pharmaceutics, becauseit is a short of research novelty. The authors described the PVCL (poly(vinyl caprolactam)) is still in its infancy, especially in comparison to PNIPAM (poly(N-isopropyl acrylamide) , but there are some similar articles about the  nanoparticles including drug for cancer therapy. I feel the authors should emphasize the novelty of this article and the system discussed in this manuscript is superior to the other nanoparticle system. Moreover, the reviewer thinks that the results are not well discussed and conclusions are not meaningful in manuscript.

The major points are described as below.

In Figure 1 legend, (c) is lost.

Considering the order of the structure of the text, Figure 2(d) should be renumbered as Figure 2(a).

Please check the Figure 3 legend again. Figure and description do not match.

In Figure 5 legend, (a) and (c) are lost.

Page 9, line 327, Fig 6B and C were described, but there is no Fig. 6!

Did the authors have the release profiles of DOX at 5 and/or 10 mM DTT  at 25 degree C? The reviewer thinks the faster release profiles 5 and/or 10 mM DTT  at 25 degree C would show rather than 37 degree C.

In vitro cytotoxicity,  the authors discussed to Cys-BIS-P(VCL-HEA) nanogels effectively inhibited the HepG2 cancer cells with the IC50 value, but cell viability seems to be almost the same between the 10 mM GTH group and DOX alone at DOX concentrations of 1 mg/mL or higher, since the tolerated dose of DOX in HepG2 cells and the actual concentration of  anticancer activity are unknown. It cannot be said that this formulation is more safety rather than Dox alone from this result.

In cellular uptake test, I do not recognized the more Dox fluorescence at 10 mM of GTH intensity around the nucleus and inside the nucleus than 5 mM GTH from Fig. 5(C), the authors should add a quantitative test or a test with a different cell system or fluorescent dye.

Author Response

(The authors gave the same response as above.)

Reviewer 3 Report

  1. In Line 39, the author claimed that “Only one innovative drug delivery system (DDS) has the ability to target and control 39 drug release [4].”, this is not clear and misleading, is this only one example in clinic? 
  2. The introduction section of this manuscript need to be more concise, it seems like the main focus of this paper is proposing a new redox-responsive nanogel system for drug delivery, the authors can shorten the first two paragraph to talk about the background, and give more introduction to what is lacking in nanogel-based drug delivery, and talk about what redox sensitive systems have been reported to solve these problem and then state what’s new about your system. Important papers regarding redox-responsive nanogels including a) Angew. Chemie - Int. Ed. 2020, 59 (52), 23466–23470. b) Materials Science and Engineering: C 118 (2021): 111536. c) Biomacromolecules 23 (1), 339-348; d)ACS applied materials & interfaces 10.19 (2018): 16698-16706.
  3. Figure 5b, why GSH need to be added to the cell culture to induce the release? intracellular GSH level is also mM range, if pretreated with GSH, how do you know if the drug is released extracellularly and uptake by the cells? Or the drug was released in side the cells? 

Author Response

  1. In Line 39, the author claimed that “Only one innovative drug delivery system (DDS) has the ability to target and control 39 drug release [4].”, this is not clear and misleading, is this only one example in clinic? 

Response: Thank you for your comment. Now we have revised the sentence.

  1. The introduction section of this manuscript need to be more concise, it seems like the main focus of this paper is proposing a new redox-responsive nanogel system for drug delivery, the authors can shorten the first two paragraph to talk about the background, and give more introduction to what is lacking in nanogel-based drug delivery, and talk about what redox sensitive systems have been reported to solve these problem and then state what’s new about your system. Important papers regarding redox-responsive nanogels including a) Angew. Chemie - Int. Ed. 202059 (52), 23466–23470. b) Materials Science and Engineering: C 118 (2021): 111536. c) Biomacromolecules 23 (1), 339-348; d) ACS applied materials & interfaces 10.19 (2018): 16698-16706.

Response: Thank you for your suggestion. Now we have cited the references in the introduction section [14-17].

  1. Figure 5b, why GSH need to be added to the cell culture to induce the release? intracellular GSH level is also mM range, if pretreated with GSH, how do you know if the drug is released extracellularly and uptake by the cells? Or the drug was released in side the cells? 

Response: Thank you for your suggestion. We assume that the treatment of GSH with HepG2 cells improves the redox levels in cells. We agree your comment. We have repeated experiment without GSH treatment of HepG2 cells. As per literature, HepG2 cells overexpress intracellular GSH levels. We repeat the experiment without GSH. The cytotoxicity and cell internalization study clearly demonstrated the drug was released in cancer cells.

Reviewer 4 Report

This is an interesting study about temperature and redox responsive nanogels for tunable intracellular doxorubicin delivery in cancer cells. I suggest it for publication after the following points are addressed.

  1. Scheme 1, the chemical structure of the drug is not clear due to the overlapping. The chemical structure of the copolymer is encouraged to be added in this scheme.
  2. In the introduction, the reason for using HEA as the hydrophlic part should be added.
  3. Line 79-80, 'Although many redox degradable disulfide linkers based nanocarriers have been previously reported [22-24]', several other studies (doi.org/10.1080/00914037.2020.1857382; doi.org/10.1021/acs.biomac.6b00168) using differnt disulfide linkers approaches (shell crosslinking and interlayer crosslinking) should be included.
  4. Figure 1, 3, and 5 caption doesn't fit to the figure.
  5. The resolution of figure 5 should be improved to a higher level and the text in the figure is too small.
  6. At 37 oC, it is above the LCST of PVCL, why DTT made the burst release of nanogel? Although disulfide bonds were broken down under DTT, the nanogel should form micelles if the conc. was above the cmc of copolymer. Such discussion should be added.
  7. It is not clear why temperature responsive is designed in this study.

Author Response

This is an interesting study about temperature and redox responsive nanogels for tunable intracellular doxorubicin delivery in cancer cells. I suggest it for publication after the following points are addressed.

  1. Scheme 1, the chemical structure of the drug is not clear due to the overlapping. The chemical structure of the copolymer is encouraged to be added in this scheme.

Response: Thank you for your comment. Now we have provided the chemical structure of the crosslinked nanogels structure.

  1. In the introduction, the reason for using HEA as the hydrophilic part should be added.

Response: Thank you for your comment. Now we have provided the importance of hydrophilic HEA in the introduction part.

  1. Line 79-80, 'Although many redox degradable disulfide linkers based nanocarriers have been previously reported [22-24]', several other studies (doi.org/10.1080/00914037.2020.1857382; doi.org/10.1021/acs.biomac.6b00168) using differnt disulfide linkers approaches (shell crosslinking and interlayer crosslinking) should be included.

Response: Thank you for your comment. Now we have provided important citations in the introduction part [18, 19].

  1. Figure 1, 3, and 5 caption doesn't fit to the figure.

Response: Thank you for your comment. Now we have provided the figure captions.

  1. The resolution of figure 5 should be improved to a higher level and the text in the figure is too small.

Response: Thank you for your suggestion. Now we have provided corrected Figure.

  1. At 37 oC, it is above the LCST of PVCL, why DTT made the burst release of nanogel? Although disulfide bonds were broken down under DTT, the nanogel should form micelles if the conc. was above the cmc of copolymer. Such discussion should be added.

Response: Thank you for your comment. Now we have discussed in the results and discussion part.

  1. It is not clear why temperature responsive is designed in this study.

Response: Thank you for your suggestion. Now we have provided the importance of temperature responsive in the introduction part.

Round 2

Reviewer 4 Report

It can be accepted as the current form.

Author Response

Response: Thank you for your suggestion. Now we have rechecked and corrected all errors in the manuscript.